# Synergetic Thermal Therapy for Cancer: State-of-the-Art and the Future

**DOI:** 10.3390/bioengineering9090474

**Published:** 2022-09-15

**Authors:** Qizheng Dai, Bo Cao, Shiqing Zhao, Aili Zhang

**Affiliations:** School of Biomedical Engineering, 400 Med-X Research Institute, Shanghai Jiao Tong University, 1954 Huashan Road, Shanghai 200030, China

**Keywords:** cancer, thermal therapy, combination, physical therapy

## Abstract

As a safe and minimal-invasive modality, thermal therapy has become an effective treatment in cancer treatment. Other than killing the tumor cells or destroying the tumor entirely, the thermal modality results in profound molecular, cellular and biological effects on both the targeted tissue, surrounding environments, and even the whole body, which has triggered the combination of the thermal therapy with other traditional therapies as chemotherapy and radiation therapy or new therapies like immunotherapy, gene therapy, etc. The combined treatments have shown encouraging therapeutic effects both in research and clinic. In this review, we have summarized the outcomes of the existing synergistic therapies, the underlying mechanisms that lead to these improvements, and the latest research in the past five years. Limitations and future directions of synergistic thermal therapy are also discussed.

## 1. Introduction

Thermal therapies have been studied for over thirty years. Temperature above or below the physiological temperature is used to locally treat the tumor. It is regarded to be safe, minimal-invasive, or non-invasive for tumor treatment. The process can be repeated several times without generating resistance compared to chemotherapy. The thermal energy can be well controlled in the target region through image guidance and elaborated control strategy. Based on the operative temperature, thermal therapy could be divided into three categories: thermal ablation (>55 °C), hyperthermia (>39–45 °C), and cryoablation (<−40 °C). It is becoming more popular and is widely used in clinical treatments of lung cancer [1,2], pancreatic cancer [3], anal carcinoma [4], unresectable extrahepatic cholangiocarcinoma [5], breast cancer [6], benign thyroid nodules (BTNs) [7], renal cell carcinoma (RCC) [8], and hepatocellular carcinoma (HCC) [9]. Patients recover quickly after complete local ablation and with very limited side effects.

The mechanisms of thermal therapies can be explained from the perspective of effects on cells [10,11]. In cryoablation, cells are killed by low temperature induced direct cell injury, apoptosis, vascular injury and ischemia, and immunomodulation. In thermal ablation and hyperthermia, the cell killing effect depends on the thermal dosage. Thermal damages include but are not limited to (1) cell membrane collapse, protein denaturation, mitochondrial dysfunction, halted enzyme function for over 50 °C, (2) inhibition of DNA replication, halted metabolism, the release of chemokines & cytokines for 41–45 °C, (3) release of pro-inflammatory cytokines, tumor antigens, RNA, DNA, heat shock proteins (HSPs) & high mobility group protein B1 (HMGB1) and so on. Besides, more evidence from recent studies indicates that thermal therapy could induce the immune response [12,13,14,15,16,17]. In tumor-draining lymph nodes (TDLN), thermal therapy can double the expression of CCL21 to make up for the reduced T cell homing [18]. Thermal therapy induces HSPs to modulate the immune system, including improving dendritic cells and natural killer cells’ function, mediating lymphocyte-endothelial adhesion and leukocyte trafficking [19]. In addition, the favorable immune microenvironment in TDLN and organs creates hope for a better outcome.

Due to its profound effects on the tumor (at different levels from molecular to global), thermal therapy is applied together with other cancer treatment modalities either in sequence or at the same time. The combination of therapies has shown great potential for synergism. In synergetic therapies, the heating/cooling effects alter the physical, chemical, and physiological characteristics of the tumor, which either improves the efficiency of the other modality or decreases the required dose of thermal energy. Understanding how the combined therapies work would help design new cancer therapy modalities while ensuring better treatment outcomes.

In this review, the state-of-the-art research results of synergetic thermal therapy for cancer are summarized to provide a comprehensive overview of current treatments combined with thermal therapies, their synergistic mechanisms, and a discussion of future possibilities for cancer treatments combined with thermal therapy.

## 2. Current Thermal Therapy Technique for Cancer

The heating of tissue can be realized by either electromagnetic or mechanical waves. Typical treatments include radiofrequency ablation (RFA), microwave ablation (MWA), laser interstitial thermal therapy (LITT), magnetic particle hyperthermia (MPH), photothermal therapy (PTT), and high intensity-focused ultrasound (HIFU). The wavelengths/frequencies used are listed in Table 1.

Thermal ablation uses a temperature higher than 55 °C to induce direct coagulative necrosis of the targeted tumor tissue. For RFA/MWA. under the guidance of medical imaging [20,21], the oscillating electromagnetic field is delivered to the tumor tissue through interventional probes, resulting in high lethal temperature [22]. In LITT (also called stereotactic laser ablation (SLA)), a laser probe is usually inserted stereotactically into the tumor and the light is absorbed by the tumor tissue with its temperature heated to lethal point [23,24]. For HIFU, the acoustic wave is converged to a small focus, causing a very quickly rising of the tissue temperature. Through either mechanical or electrical scanning, the whole region of the tumor can be covered [25]. In these ablation treatments, due to the thermal diffusion, the region of the tissue that surrounds the region with high lethal temperature will experience temperature that is in the range of hyperthermia (>39–45 °C).

MPH (magnetic particle hyperthermia) and PTT are effective hyperthermia treatments. For MPH, the magnetic particles serve as embedded thermal sources in the tumor when exposed to the external alternating magnetic field [26]. The generated heat spreads and covers the target region [21]. The photothermal therapy (PTT) uses the nano-particles loading with a photothermal transducer (PTA) and the laser to form a laser absorption peak thus better confining the thermal energy in the target [27,28,29,30,31]. Hyperthermic-isolated limb perfusion (HILP) [32], intraperitoneal hyperthermic perfusion (IPHP) [33], and intrapleural hyperthermic perfusion (IHP) [34] are also available hyperthermia modalities. The heating power of the above thermal ablation technologies can all be controlled to the hyperthermia range.

Cryoablation is usually realized by high-pressure gas or low-temperature liquids. The principles of freezing include the rapid expansion of argon, nitrogen, and nitrous oxide through the Joule-Thomson effect [41] and phase change of liquid nitrogen. Freezing is induced in the tumor tissue through a tiny probe. The freezing-induced damage depends on the lowest temperature, cooling rate, the time kept at the lowest temperature, thawing rate, and freezing cycles.

For now, the application of heating is more common than cooling in the clinic. This may be due to the fact that the control of electromagnetic waves and mechanical waves is much easier and more stable than that of freezing liquids or gases [42,43]. Unlike heating, the cooling of the same target may be repeated 2–3 times [44] which increases the treatment duration. However, as the freezing-induced anesthetic has an effect and blood perfusion slows down, the patients may undergo less pain and the influence of blood flow may not be as significant as in heating. With better development of control techniques in cryosurgery, the application of cryosurgery may become more popular.

## 3. Cancer Treatments Combined with Thermal Therapy

The above clinical thermal therapy techniques have been reported to show promising results from either in vitro studies or clinical retrospective evaluations in combination with other tumor treatment modalities, which include chemotherapy, radiation therapy, immunotherapy, and other minimally invasive therapy (Figure 1).

### 3.1. Chemotherapy Combined with Thermal Therapy

Chemotherapy is a major cancer treatment. Chemotherapeutic agents such as brivanib, doxorubicin (DOX), erlotinib, everolimus, lenvatinib, linifanib, ramucirumab, regorafenib, sorafenib, sunitinib, tivantinib [45] are delivered by oral protocol, bolus injection, and continuous infusion to circulate in the whole body [46]. These agents inhibit the kinases and key factors related to cell proliferation and thus kill the cancer cells. Local thermal treatments have already been shown to be able to increase chemotherapy outcomes in clinical applications, but the enhancement varies with different kinds of cancers. In lung cancer (ICC), the medium survival (95% CI)-OS of ablation-chemotherapy (RFA/MWA as thermal therapy) almost doubled compared to that in the chemotherapy- only group after propensity score matching (PSM) [47]. When combined with chemotherapy in lung cancer, MWA/RFA increases median PFS time up to 10.4/9.2 months [48] respectively. In ovarian cancer liver metastasis (OCLM), the 1-, 2-, and 3-year OS rates of RFA plus chemotherapy were 93.3%, 80.0%, and 53.3% respectively, compared to 79.5%, 60.1%, and 42.1% in the chemotherapy alone [49]. The mechanism of synergetic thermal therapy and chemotherapy is summarized in Figure 2.

The performance of thermally enhanced chemotherapy can be summarized as increased DNA damage and chemotherapeutic drug targeting, inhibition of DNA repair, formation of an acidic environment, and increased sensitivity to chemotherapeutic agents especially for cells in the S phase [50,51,52]. Dou, J. et al. [53] found that the elevated intracellular Ca^2+^ caused by combining MWA and DOX destroyed the homeostasis of tumor cells and decreased the mitochondrial inner membrane potential, resulting in massive apoptosis in vitro studies on HepG2 cells and in vivo studies on mice.

When combined, the dosage thresholds of either chemotherapeutic agents or thermal energy either hyperthermia or thermal ablation can be reduced. In treating DU145 prostate cancer cells, the combination of HIFU and Sorafenib decreases the thresholds by almost 67% and above 80% respectively compared to HIFU alone and Sorafenib alone according to cell viability [54].

Increase in cryotherapy efficiency has also been observed when used with chemotherapy [55,56]. Application of cisplatin increased lethal cooling temperature for bladder cancer from −25 °C to −15 °C [55]. With 30 min of 1 μM cisplatin exposure before cooling to −15 °C, the killing effect based on the survival rate of SCaBER cells was increased 4.6 times compared to the group with freezing alone. The low dosage of the chemotherapeutic agents used is not enough to kill the cells directly but is believed to have prevented the initiation of cellular repair mechanisms necessary for surviva1 [55], substantially activating caspase-3 within the nucleus to induce apoptosis [56] or increase the Bcl-2 to Bax ratio leading to a pro-death tendency [57], thus increases the low-temperature induced damage.

When the chemotherapeutic agents are packed as nano-drugs, the spatial distribution can be further controlled and targeted to decrease the byproduct of chemotherapy by application of thermal energy [58] with therapeutic effect enhanced and side effects decreased from the followed aspects.

(1) Local heating is used to break the structure of the nano-particles and release the chemotherapeutic agents [59,60,61,62]. Only after being heated to a certain temperature, the temperature-sensitive liposomes will release the chemotherapeutic agents [54]. A newly designed microneedle consisting of photothermal agents and DOX is used to realize a NIR-II light-triggered heating, local drug release, and NIR-II fluorescence imaging at the same time [39]. Furthermore, through a programmable and wireless control of heating duration, the amount of DOX released from the temperature-sensitive liposomes is successfully controlled [63]. Temperature sensitive nano-particles can also be made from different materials such as Evans blue derivative-functionalized gold nanorods [64], CoFe_2_O_4_@PDA@ZIF-8 sandwich nanocomposite [65], DNA-templated silver nanoclusters and polydopamine nanoparticles [66], camptothecin-conjugated gold nanorods [67], and ZIF-8 coated ZrO_2_ [68] for thermally controlled release of the drug. These particles are also found to be able to increase local adsorption of heat at the same time. And for PTT chemotherapeutic-drug loaded with photothermal transduction agents (PTAs) have shown higher photothermal conversion efficiency and increased penetration of PTT. Combining the functional effects of PTT-mediated thermal effects, such as controlled drug release and enhanced chemical response of targeted tissues, synergistic strategies can achieve a cure of larger tumors as well as lesions with distant metastatic and disseminated.

(2) Heating enhances vascular permeability, increases oxygenation, decreases interstitial fluid pressure, reestablishes normal physiological pH conditions, thus increases extravasation of the nano-drugs from the bloodstream and cellular uptake of the chemotherapeutic agents, and, finally, enhances the killing effect [69,70].

(3) Targeting of the thermal therapy to the tumor is improved by the nano-drugs designed [66,71]. When nano-drugs in the cancer tissue serve as multiple thermal sources, the special distribution of the nano-drugs generates a unique heating pattern and thus helps further focus the application of thermal energy [72]. Besides, by surface modification, the nano-particles would target the cell surface [58,73] or even subcellular structures such as mitochondria [74] and thus realizing heating targeted region accordingly. Also. by using materials with different thermal, conductivities, the temperature distribution inside the tissue may be changed accordingly to the increased thermal energy accumulation in the targeted tissue.

Due to the above synergistic effect of thermal energy and nano-drugs, the chemotherapeutic byproduct effect is greatly reduced [75,76] with local killing effects significantly increased [54,55]. Through further precise local and temporal control of both drug and thermal energy, the combination of thermal energy and nano drug technique is becoming one of the most promising directions for cancer treatment.

### 3.2. Radiation Therapy Combined with Thermal Therapy

Radiation therapy uses high-energy photon radiation such as X-rays, and gamma (γ)-rays to generate ROS and induce single-strand breaks (SSB) and double-strand breaks (DSB) in DNA to terminate cell division and proliferation [77]. The combination of thermal therapy and radiation therapy can date back to 1975 [78]. Heating to 45 °C several minutes before/during radiation effectively radio-sensitized the radio-resistant G1/S phase cells. Thermally enhanced radiation therapy has also been found to alleviate the hypoxia degree of tumor and a decrease in the expression of DNA repair-related proteins [79]. The mechanism of synergetic thermal therapy and radiation therapy is summarized in Figure 3.

Improved local control in spinal metastases has been found by combined thermal ablation of hyperthermia and radiation compared to hyperthermia alone [80]. With the development of non-invasive hyperthermia in 2018, where a 12 antenna applicator for targeted selective heating [81] was designed, the heating was focused into the tissue deep-seated in the body for advanced head and neck carcinoma 1–3 h after irradiation. The maximum temperature and median temperature can reach 42.3 °C and 39 °C respectively. The response rates after 3 months were 46% (complete) and 7% (partial) and no severe complications or thermal toxicities were observed. This study provided the possibility for synchronous heating and radiation though the procedure of combined therapy was made by alternating heating and radiation. Photothermal therapy (PTT) is another hyperthermia modality used before or after radiation therapy [82,83,84]. Other than the heating effects, the metal ions of the photosensitizer strongly absorb, scatter, and re-emit radiation energy, and thus generating extra singlet oxygen to amplify the local radiation dose [85].

Thermal ablation is also used together with radiation. With developed in vivo pancreas models in swine, the scores of the injury to target in radiation therapy applied within 12 h followed HIFU group was found to be almost 2–3 times of the HIFU only group and about 1.5 times to the radiation therapy only group [86]. Large areas with reactive/swollen cells were observed by H&E staining. Radiation would help ensure the complete killing of the cells located at the outer ring of the carcinoma which is sensitive to radiotherapy but may escape from the thermal energy due to large cooling from blood perfusion in this region.

In these cases, thermal therapy and radiation therapy are still applied using separate machines. The time duration between the joint treatments sometimes is quite long, where the synergistic effect of both treatments found through in vitro experiments hardly exists in these clinical trials. The integration of synchronous thermal ablation/hyperthermia with radiation therapy needs further study to realize a better design of the treatment sequences and even better treatment outcomes than current findings.

### 3.3. Immunotherapy Combined with Thermal Therapy

Cancer immunotherapy is becoming more and more popular in recent years. The target of immunotherapy lies in the excitation of a strong global systematic immune response against cancer, especially the metastasis and invisible micro-metastasis that may escape from current treatments. Immune checkpoint blockade therapy, especially PD-1/PD-L1 and CTLA4 antibodies, and CAR-T therapy are discussed as representative forms of immunotherapy due to the rapid development and trials in combination with thermal therapy.

#### 3.3.1. Immune Checkpoint Blockade Therapy Combined with Thermal Therapy

Immune checkpoint inhibitors (ICIs) such as programmed cell death protein 1 (PD-1)/programmed death-ligand 1 (PD-L1) antibodies and cytotoxic T-lymphocyte-associated protein 4 (CTLA-4) antibodies have been approved by the FDA for treatment of malignant melanoma, lung cancer, and lymphoma [87]. The immune checkpoint activity is inhibited to counteract the suppressive effect of the tumor microenvironment on the host’s immune activity, and thus reactivating the T-cell-based antitumor immune response. Therefore, an adequate and effective immune response is critical for successful oncology treatment. However, insufficient tumor antigenicity, lack of lymphocytes, and immunosuppressive tumor microenvironment often obstruct this therapy [87]. The immune-related adverse events (irAEs) [88] have been also widely observed clinically, including cardiotoxicity [89,90], pancreatic injury [91], thyroid Dysfunction [92], and infectious complications [93,94].

Thermal therapy itself has been reported to result in certain anti-tumor immune effects but not strong enough for metastases treatment [10,95]. A combination of thermal therapy and immunotherapy [96,97] has been studied to realize a positive immune response. The medium recurrence-free survival (RFS) increased from 19.3 weeks to 39.1 weeks when the PD-1/PD-L1 antibodies were added to RFA, while the OS increased slightly from 47.6 weeks to 51.0 weeks in a retrospective study of 127 patients with HCC [98]. In renal cell carcinoma (RCC) [99] and colorectal carcinoma [100], the combination of RFA/MWA and immune checkpoint inhibitors (ICIs) also showed potential in complete local control, prevention of the growth of metastases, and significant increase of the patient’s survival rate.

These results are very promising and many studies have looked into the mechanisms underlying the joint treatment. RFA and MWA have been proven to serve as antigen sources for immunotherapy in B16-OVA tumor-bearing mice [101]. The induction of heat shock proteins, acute phase response, and mobilization of antigen-presenting cells and effector lymphocytes after local thermal ablation may have enhanced the immune response from the ICI treatment [102]. Similarly, a couple of studies have been proposed to combine HIFU with immunotherapy. The advantages of such combination can be concluded as: (1) Thermal therapy induced increase of IFN-γ, HSP 27, HSP 70 concentrations and decrease of IL-10, IL-4, TGFb-1, and TGFb-2 concentrations to help improve the antigen-presenting capability [16,103,104]; (2) Combined treatments also resulted in increased concentrations of dendritic cells and decreased concentrations of IL-10 and CD4${}^{+}$Foxp3${}^{+}$, leading to more efficient priming and activation of T-cells [13,105]; (3) Combined therapy induced elevated long-term memory markers CD4${}^{+}$CD44${}^{+hi}$CD62${}^{+low}$ and CD8$\alpha^{+}$CD44${}^{+hi}$CD62${}^{+low}$ for prevention of tumor recurrence and thus enhanced sensitization of ICIs therapy [106]. In addition, the mechanical effect of HIFU also plays a significant part in improving the anti-tumor immune response, as the addition of the microbubbles was found to have induced the Th1 reaction to strengthen the activity of DC and cytotoxic lymphocytes [107].

#### 3.3.2. CAR-T Therapy Combined with Thermal Therapy

CAR-T cell therapy ( Chimeric Antigen Receptor T-Cell) is a T cell-based therapy, which combines the technologies of immunotherapy and gene therapy [108]. It first introduces a gene to the person’s T cells to modify the immune cells to attack cancer cells. In recent years, CAR-T cell therapy combining thermal therapy is receiving a lot of attention from researchers. Local thermal therapy is able to modulate the expression of the heat shock promoter through temperature control [109,110], allowing induction of the associated transgene only in heated areas. The mechanism of synergetic thermal therapy and CAR-T therapy is summarized in Figure 4.

Because of its precise and rapid heating properties, focused ultrasound(FUS) has a great application for the control of transgene expression in oncotherapy. Wu, Yiqian et al. [109] have achieved specific activation and direct control of CAR-T cells without any exogenous cofactor, suppressing tumor growth apparently in vivo. The heat-shock-protein promoter they used can sensitively control the expression state of chimeric antigen receptor(CAR) under the simulation of FUS, avoiding the problem of off-target tumor toxicity associated with CAR-T cell therapy. In addition, Miller, Ian C et al. [110] achieved the enhanced intratumoral activity of CAR-T cells under photothermal control. In vitro experiments showed that transgene had over 60-fold-higher expression without any influence on normal physiological activities under mild temperature elevations(to 40–42 °C) for 15–30 min. While in vivo study found that photothermal conversion of gold nanorods stimulated heat-shock elements(HSEs) and core promoters to dive IL-15 superagonist activation, greatly enhancing the anti-tumor capacity of CAR-T therapy.

#### 3.3.3. Other Immunotherapies Combined with Thermal Therapy

Compared to thermal ablation techniques, it is believed that cryoablation preserves more intact tumor antigenic structures to trigger a tumor-specific protective immune response [10,111]. Shao qi et al. have demonstrated that cryoablation (−80 °C, 30 min) on B16 cancer cells resulted in an 8.25 times release of natural proteins and activated a stronger T-cell immune response, which is 1.6 times more than the heating treatment group (50 °C, 30 min) [112]. Combining cryoablation, IFN-γ and VEGF signaling pathways are activated and further enhanced, providing a good induction pathway for immune cell infiltration in the tumor microenvironment [113]. In addition to the ability to increase the killing activity of immune cells such as T cells and NK cells, the underlying mechanism may contribute to the modulation of immune responses by promoting a release of Hsp70 and reducing Ki67 activity [114,115,116]. Recently, Campbell, M.T. et al. conducted a preliminary study using Tremelimumab immunotherapy with and without cryoablation in 29 patients with metastatic renal cell carcinoma [113]. They found a significant increase in T-cell infiltration in the tumor microenvironment of patients after cryo-trimethoprim combination therapy and laid an important foundation for subsequent combination therapy for patients with mccRCC. Except for the usage of ICIs and CAR-T therapy, some other immunotherapies such as the usage of toll-like receptor (TLR) agonists, adoptive cell therapies (ACTs), and uptake of epigenetic modulators can also be combined with cryoablation to enhance its immunogenicity, and thus stimulate the immune system and lead to good synergistic immune-mediating effects [117,118,119,120].

### 3.4. Gene Therapy Combined with Thermal Therapy

Even since the concept of gene therapy was first put forward by Friedmann in 1972 [121], it has become an important modality for cancer therapy. This technology can treat cancer at its genetic root by editing a gene to enhance the efficacy of target cells, significantly improving the survival rate of cancer patients [122]. However, the major problem of this therapy is the accurate delivery of the gene expression vectors and the effective expression of target genes without affecting the cells’ normal regulatory activities [123]. As common gene therapy vectors, Adenovirus, lentivirus, and adeno-associated viruses (AAVs) gene therapy vectors have potential immunostimulatory effects, lower targeting and oncogenic effects [124,125,126]. Non-viral delivery vehicles such as liposomes, nanoparticles, polymers, and dendrimers have low transfer efficiency, and most of the transduced genes in this therapy are metabolized and eliminated by the liver and kidneys [127,128].

In earlier studies, researchers have demonstrated that heating can rupture tissue, and improve cell metabolism & cell membrane permeability, thereby increasing the efficiency of gene transfer [129,130]. The mechanism of synergetic thermal therapy and gene therapy is summarized in Figure 5. Du et al. [131] developed an MRI-heating guidewire for enhancing vascular gene transduction and expression. In this research, the heated guidewire is capable of transferring external RF energy into the vasculature, and the cylindrical temperature distribution acts precisely on the target vascular area to increase the delivery efficiency. In addition, researchers also have investigated a variety of thermal methods to enhance gene delivery in vivo, including RFA [131], HIFU [132,133] and thermally Responsive Polymers [134].

Results of these studies have shown that through precise control of the thermal parameters, control of genetic engineering for therapeutics can be achieved [109]. The combining of thermal and gene therapy for cancer treatment will help lay an important foundation for the development of safer and more controllable application of gene therapy in clinics [135].

### 3.5. Other Minimally Invasive Therapy Combined with Thermal Therapy

Other than the above therapies that can be greatly enhanced by thermal therapy, some other local minimally invasive treatments such as TACE [136], TARE [137,138], and IRE [139,140] have also been used together with the thermal therapies. RFA has been shown to increase the OS and progression-free survival (PFS) of unresectable pancreatic cancer [141], HCC [142,143,144] and ICC [145] treated with TACE. Disruption of the angiogenesis and blood supply of the tumor enhanced the therapeutic outcome of TACE [136]. The combination of cryoablation and TACE has resulted in a much-reduced blood supply to the tumor than any single modality. Similarly, transarterial radioembolization (TARE) has shown a highly increased radiation effect when in combination with thermal therapy. By lowering the lethal electric field threshold for Irreversible Electroporation(IRE) [139,140] by heating from RFA, the therapeutic outcome of IRE was obviously increased.

## 4. Combined Multi-Therapies

Combined multiple thermal therapies are showing up in recent years, bringing brand new discoveries. Greatly enhanced immune effects have been discovered through combined cryoablation and thermal therapy [69,146,147,148,149]. Such combined therapy could create an inflammatory environment for stimulation of T cell activation and differentiation, and thus promoting long-term antitumor immunity through inducing the release of tumor-related DAMPs, especially tumor-related HSP70 in situ and the peripheral [150], down-regulating TNF-α levels [151], and releasing iron [152]. The synergetic effects are also shown as induced tumor-related DAMPs improving the thermal sensitivity and decreasing the lethal dosage of thermal therapy. Hai Wang et al. [153] combined cryoablation, photothermal heating, and nano-drug delivery for the treatment of orthotopic triple-negative human mammary cancer with promising results by both in vitro and in vivo studies.

Thermal therapy-assisted micro-robots or nano-robots are also newly developed techniques. Compared to the nano-particles, micro-robots or nano-robots can home in tumors actively instead of passively gathering. Kalyani Ektate et al. [154] developed “thermobots” which actively transported low-temperature sensitive liposomes (LTSLs) to the tumor and work with HIFU. [155]. Furthermore, micro-robots might be used to perform local precise radiation therapy or chemotherapy and may also be used together with thermal therapy for the design of more effective solutions for patients.

Another core of multi-therapy is immunotherapy. The synergetic thermal therapy with other treatments can be used together with immunotherapy [156]. Qiang Ji et al. [157] proposed to combine thermal ablation with both TACE and immunotherapy. It was found that in the RFA+TACE+CIK group, the levels of CD3${}^{+}$ T cells, CD3${}^{+}$CD4${}^{+}$ T cells, CD4${}^{+}$/CD8${}^{+}$ ratio, Treg and NK cells were significantly higher compared to the RFA+TACE group. The introduction of immunotherapy into the current synergetic thermal therapy may finally control the potential distant metastasis and prevent any possible recurrence of the tumor. The alternated freezing and RF heating therapy was also used to work together with immunotherapy. Block of tumor immunosuppressive factor CCL5 has been combined with alternated freezing and RF heating to promote Th1 differentiation and the cytotoxicity of CD8${}^{+}$ T cells to improve survival rate by down-regulating the proportion of MDSCs and induced full M1 macrophage polarization in 4T1 murine breast cancer [158].

## 5. Discussion

A review of recent works on thermally combined cancer therapies has shown positive progress in joint treatments. Multi-level biological responses to thermal therapy including alternation in the physical and chemical environment of the tumor, direct or indirect cancer cell death, change of cell status, vascular damage, and stimulation of the systemic inflammatory reaction are the major foundations for the joint treatments. The synergetic effects achieved include a decrease of both dosages of both therapies and overcoming the shortcomings of every single modality. The delivery efficiency of the gene vectors and chemotherapeutic drugs can be increased by thermal therapy while the nano-drugs used for chemotherapy have also been used to increase the delivery and accumulation of thermal energy. These positive results have significantly enhanced the therapeutic effect of the tumor. Cancers with strong resistance to conventional therapies, with no target on the cell surface for immune therapy, or being highly aggressive might benefit from the thermally combined treatment.

Though synergetic thermal therapy has received many breakthroughs, there are still some limitations in current synergetic thermal therapies. Key points that need to be considered for future developments of synergetic thermal therapy include:

(1) The dosage used in current studies are manually and empirically decided, further systemic optimization of the combined dosage of both combined modalities is needed.

(2) The treatments are operated in different clinical departments. A platform offering different thermal therapy choices to be combined with any other modalities may be a possible solution for the wider application of synergetic therapies.

(3) To achieve better treatment outcomes and more wide application, delivery of the thermal energy shall be more minimal-invasive or non-invasive. The spatial control precision of thermal energy needs to be further improved to target special cells and even subcellular structures [159]. The Sono-magnetic heating, which uses a magnetic field and focused ultrasounds (FUS) in the presence of iron oxide nanoparticles provides a choice of precise temperature regulation [160].

(4) Selection of the most suitable combined thermal therapy remains an issue to be solved as there are a couple of different forms of thermal therapy that can be used for combination. Existing rules for selecting certain single modalities based on the physical, chemical, and geometrical characteristics of the cancer cases may still be suitable for the choice of combined thermal therapies. However, finding the best-combined solution for specific cancer cases remains a challenge as so many choices are available and there is no direct statistical comparison of these treatments. Sometimes, if not properly selected, there could be an inhibition effect from each other [161].

(5) For thermally combined immunotherapy, which may be a big breakthrough in future tumor treatment, there is still no clear quantified relationship between the dosage of either modality in combined therapy and the degree of the anti-tumor immune response. The causes and mechanisms of systemic immune responses induced by combined therapy also remain to be answered.

(6) There are still big gaps existing between the preclinical and clinical research of thermally combined therapies. Most findings are based on cell and animal models. Whether these therapies work for humans, especially for patients who have a complex treatment history, is still an unanswered question. Sufficient clinical trials, optimized dosage planning, and real-time intraoperative monitoring of the combined treatment are necessary.

In the future, (1) more joint treatments may be developed as better treatment outcome has been achieved when more therapies are used jointly for cancer, as shown from the above review. (2) The mechanisms of synergetic effects with chemotherapeutic agents, modification of genes, and nanomaterials in thermal fields are also worth further profound study, especially at the molecular level, which would help decide the most suitable cancer for different combined therapies and even regulate the biological processes. (3) Non-thermal effects on synergetic therapy resulting from the heating techniques including electromagnetic waves, mechanical waves, et al., may help with the proposal of new applications. (4) Thermal energy may be used to enhance the precision of tumor imaging during the combined treatment. A sub-therapeutic sound pulse of HIFU for obtaining the properties of the target has shown the potential for precise guidance for thermal therapy [162]. Integration of the diagnosis system together with the thermal treatment system may be another solution for precise combined treatment. A much wider application of thermal therapy might appear in the future.

## Figures and Tables

**Figure 1 bioengineering-09-00474-f001:**
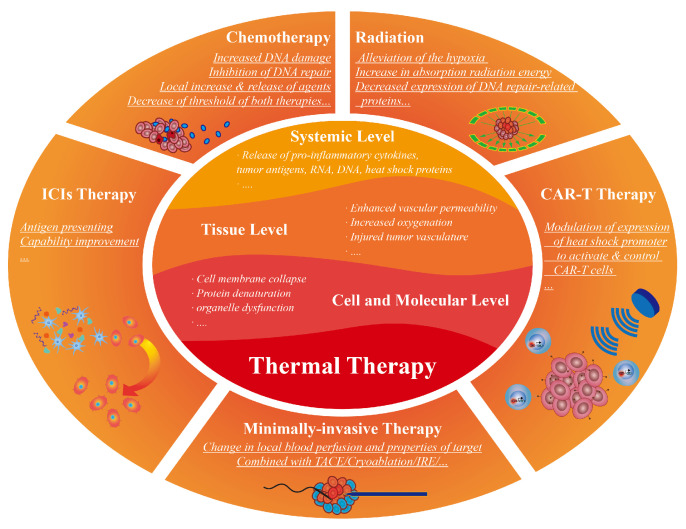
The schematic diagram of synergetic thermal therapies. The effects of thermal therapy can be divided into the cell and molecular level, tissue level, and systemic level. Due to these effects, thermal therapy is combined with chemotherapy, radiation therapy, immunotherapy, and other minimally-invasive therapy, realizing unique enhancements respectively.

**Figure 2 bioengineering-09-00474-f002:**
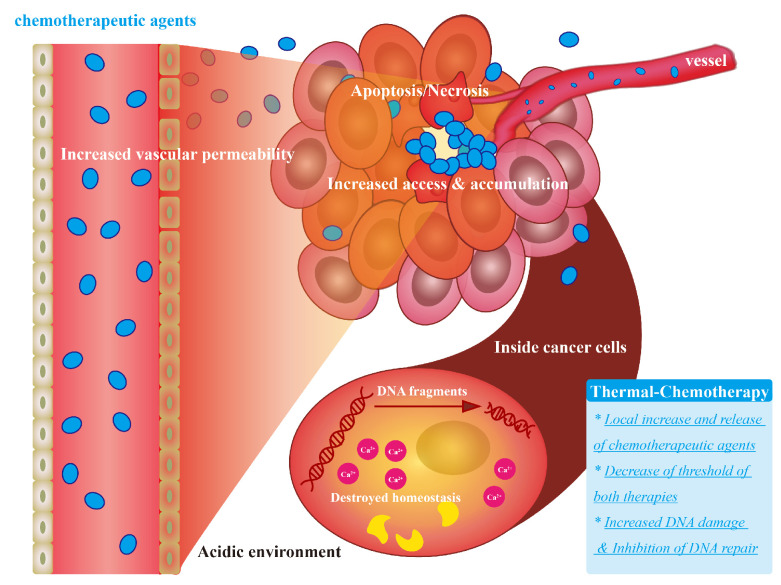
The mechanism of synergetic thermal therapy and chemotherapy. Other than induced cellular apoptosis and necrosis, thermal energy increased the tumor vascular permeability, thus increasing delivery of the chemotherapeutic agents in the heated region and increasing local drug accumulation. Heating also changes the homeostasis of cancer cells, increases drug-induced DNA damage, and inhibits the repair of DNA, which lead to the decreased lethal threshold of both modalities.

**Figure 3 bioengineering-09-00474-f003:**
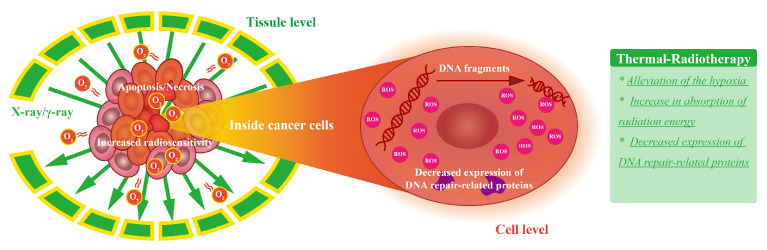
The mechanism of synergetic thermal therapy and radiation therapy. After heating to 45 °C several minutes before/during radiation, the radio-resistant of cancer cells are sensitized and the hypoxia degree of tumor is alleviated by the recruitment of oxygen. Thus, (1) increased damage to DNA by direct radiation or generated ROS is achieved, and (2) decreased expression of DNA repair-related proteins is realized. These two factors lead to the enhanced killing of cancer cells.

**Figure 4 bioengineering-09-00474-f004:**
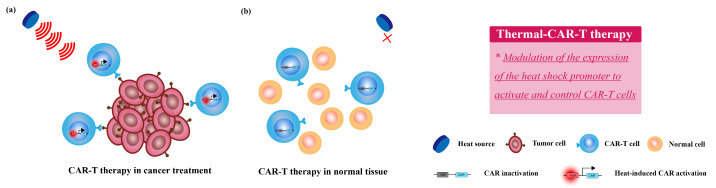
The mechanism of synergetic thermal therapy and CAR-T therapy. (**a**) T cells with inducible CAR cassettes containing heat-shock promoters are activated when heated locally and thus eradicate the tumor cells. (**b**) In normal tissue, without being heated, the T-cells with inducible CAR cassette remain inactivated and thus reducing the on-target off-tumor toxicity of standard CAR-T therapy.

**Figure 5 bioengineering-09-00474-f005:**
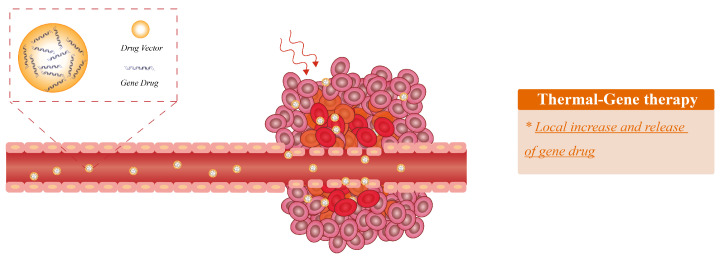
The mechanism of synergetic thermal therapy and gene therapy. Thermal therapy improves cell membrane permeability to increase the efficiency of gene transfer, achieving local increase and release of gene drugs.

**Table 1 bioengineering-09-00474-t001:** The range of wavelength/frequency applied in several different thermal therapies.

Thermal Therapy	Thermal Source	Wavelength/Frequency
RFA	Electromagnetic Field	200 kHz to 1200 kHz [35]
MWA	Electromagnetic Field	300 MHz to 300 GHz [36]
MPH	Magnetic Field	0.1 kHz to 4000 kHz [26]
LITT/PTT	Light	532 nm to 2100 nm [37,38,39]
HIFU	Ultrasound	0.5 MHz to 10 MHz [40]

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
