# Peer review of "Synergetic Thermal Therapy for Cancer: State-of-the-Art and the Future"

_bioengineering, 2022, doi:10.3390/bioengineering9090474_

Round 1
Reviewer 1 Report
In the present article, authors reviewed and summarized the literature Synergetic Thermal Therapy for cancer treatment. This is a potentially interesting review and helpful to understand the current stage of thermal therapy and to develop new strategies for future imaging applications. Some detailed comments are listed as follows.
1. What are the key points to consider to further improve the synergetic thermal therapy platforms in future developments? May authors can offer a detailed viewpoint in the end.
2. Authors should discuss limitations and solutions of PTT (ref: https://doi.org/10.1002/advs.202002504).
3. In introduction section, cite the following recent references on theranostics platforms
(https://doi.org/10.1021/acsanm.2c00170, https://doi.org/10.1039/D1BM00631B)
Reviewer 2 Report
The review by Dai et al. titled “Synergetic Thermal Therapy for Cancer: State-of-the-Art and the Future” has a good coverage of combinatory approaches of thermal therapy and conventional cancer therapy. The reviewer is impressed by the inclusion of large numbers of preclinical and clinical examples demonstrating the improved therapeutic effect of incorporating thermal therapy into existing stand-of-care. The article also covers the latest development of novel therapies.
However, the reviewer does not recommend acceptance for publication in current form, given several significant concerns and flaws. To be specific:
1. Missing critical information.
The title describes “Synergetic”. However, the word “synergy” describes: a mutually advantageous conjunction or compatibility of distinct business participants or elements (such as resources or efforts). Throughout this review, most of the arguments (and their examples) are on the side of improving conventional therapy by thermal therapy, with every little coverage on the other side on how conventional cancer therapy would benefit thermal therapy.
https://www.merriam-webster.com/dictionary/synergy
2. Biased towards hyperthermia, little coverage on thermal ablation.
Even though the review describes the difference in the introduction, the biophysics between these two thermal implementations are very different. In this review, most of the examples are given in hyperthermia, of which large scale of tissue destruction is intentionally avoided. It would be biased to focus on most of the mechanisms on hyperthermia while given “thermal therapy” in the title and abstract, with so little coverage on thermal ablative techniques.
3. Lack of original contribution on “the underlying mechanism that leads to these improvements”
Even though the authors claim to cover extensively on “underlying mechanism”, most of the concepts of combination were proposed much earlier than 5 years ago. The reviewer fails to identify any newly discovered mechanism: the review largely describes what has been well established decades ago.
There are numerous articles that have covered the benefit of thermal therapy for chemotherapy, radiation therapy and immunotherapy already, a few examples.
· Wust, Peter, et al. "Hyperthermia in combined treatment of cancer." The lancet oncology 3.8 (2002): 487-497.
· Issels, Rolf D. "Hyperthermia adds to chemotherapy." European journal of cancer 44.17 (2008): 2546-2554.
· Skitzki, Joseph J., Elizabeth A. Repasky, and Sharon S. Evans. "Hyperthermia as an immunotherapy strategy for cancer." Current opinion in investigational drugs (London, England: 2000) 10.6 (2009): 550.
4. Missing insightful perspective toward the future
In the authors’ own words, “Some future directions of synergistic thermal therapy are also discussed.” However, the reviewer sees this article only scratches the surface.
Fundamental questions retain to be answered, and the discussions by the authors do not substantially answer them, for example but not limited to:
· What disease (type of cancer, condition) can benefit from combination thermal therapy and why?
· How conventional therapy should be prescribed, given potential improvement by thermal therapy?
· How should thermal energy be delivered?
· How to control (choice and implement) thermal therapy to optimize treatment outcome?
· What aspects of thermal therapy remain to be answered?
· What are the gaps between preclinical clinical research?
Other concerns and suggestions
1. Other forms of cancer therapy were not described in this review, including but not limited to: targeted therapy, surgery, hormone therapy
2. Words in figure 1 are hard to read.
3. The concept of “Minimally Invasive Therapy” is poorly defined.
4. The reviewer does not agree that CAR-T cell therapy is a form of gene therapy.
5. “Radiation therapy” is more commonly used and more formal than “Radiotherapy”
6. Different forms of immunotherapy have distinctly different mechanisms. Immune checkpoint blockade therapy is only one of them.
Reviewer 3 Report
Many reports in the literature have shown the possibility of combining various therapies aimed at achieving a synergy effect. The manuscript presents selected achievements in the field of synergetic thermal therapy. The synergistic effect of thermal therapy is very interesting, but the authors, unfortunately, described it very briefly.
There is not a detailed, critical discussion about the result and perspective. There is no discussion of which synergetic methods are the most promising and worth researching.
The title suggests "Synergetic Thermal Therapy" but the possibility of combining two methods of heating has been omitted. For example sono-magnetic heating https://doi.org/10.1016/j.jmmm.2020.166396
The authors do not answer the question: How to control or monitor the thermal therapy process and efficiency of the synergetic process?
Table 1: Does magnetic hyperthermia use an electromagnetic field? What is the thermal source for a laser?
Figure 1: Unreadable inscriptions - fonts are too small.
Round 2
Reviewer 2 Report
The reviewer appreciates the efforts by the authors in addressing the comments. However, reading the current version of the manuscript, some of the issues remain.
To be specific:
1. Section 5 discussion seems to be hastily assembled without a clear outline. The reviewer finds it difficult to follow. The topics are not well organized.
2. A good number of the claims are unsubstantiated without reference, throughout the manuscript. Section 5 is the worse of this shortcoming. A few examples.
· It can effectively circumvent the potential toxic effects of CAR-T cell therapy in normal tissues.
· The combination of cryoablation and TACE is also valuable, which can significantly reduce the blood perfusion rate in the tissue with vascular damage.
· thus by cominbing with conventional treatmens won’t add much to the patient’s burden.
There are numerous of these instances throughout this manuscript.
3. The comments by Reviewer #1 are not fully represented in this latest manuscript, especially the questions on “What are the key points to consider to further improve the synergetic thermal therapy platforms in future developments?”. The reviewer didn’t find the contents in the response letter incorporated into the manuscript.
4. The comments from Reviewer #2 are not fully addressed, especially the concern over “Missing insightful perspective toward the future”. The authors are using languages like “more research is needed” to dodge these important questions researchers are waiting for, contrary to the promise in the abstract that “Limitations and future directions of the synergistic thermal therapy are also discussed”.
As a researcher working in the similar field, the reviewer is fully aware of most of the limitations described by the authors. However, what are the “future direction” that the authors can offer, other than just addressing the limitations?
5. The authors are still mixing the concepts of “hyperthermia” and “thermal ablation”. These 2 approaches have very different mechanisms of actions. How they differ in synergistic effect for conventional cancer therapy is not well described. For example, HYPERcollar system is a hyperthermia system, not RFA.
6. The section of “3.3. Immunotherapy Combined with Thermal Therapy” has a good number of examples, which are well covered in other related reviews. However, the mechanism of how the combination therapy would perform better is not well discussed, falling short of the promise in the abstract of “the underlying mechanism that leads to these improvements”. The reviewer would recommend reducing the length of coverage of examples (as they can be easily found in numerous reviews) and describe more on the mechanism of which other researchers would find more useful.
7. The mechanism on improving the ICI therapy is not accurate. The reviewer did consult a colleague working on this subject and find a few statements not scientifically supported. For example, in Figure 1, the reviewer does not agree that “Seeds for amplification of immune response” and “trigger sensitization of checkpoint inhibitor”, as the reviewer has no idea what they actually mean.
8. The reviewer is still confused by Figure 4: The mechanism of synergetic thermal therapy and CAR-T therapy. What are the claims from?
9. A good number of sentences are very difficult to understand. Some examples:
· When used combinedly with chemotherapy, the non-thermal effects of RFA and MWA should be crucial to decide which one is more suitable
10. The language still needs improvement. Grammar and spelling mistakes are common.
Reviewer 3 Report
In my opinion, the information in Table 1 is still incorrect. The word "laser" should be replaced by "light". For MPH the word "Electromagnetic Field" should be replaced by "Magnetic Field".
